# Identifying Current Feelings of Mild and Moderate to High Depression in Young, Healthy Individuals Using Gait and Balance: An Exploratory Study

**DOI:** 10.3390/s23146624

**Published:** 2023-07-23

**Authors:** Ali Boolani, Allison H. Gruber, Ahmed Ali Torad, Andreas Stamatis

**Affiliations:** 1Honors Department, Clarkson University, Potsdam, NY 13699, USA; 2Department of Kinesiology, Indiana University, Bloomington, IN 47405, USA; ahgruber@indiana.edu; 3Faculty of Physical Therapy, Kafrelsheik University, Kafr El Sheik 33516, Egypt; ahmed.ali.torad@gmail.com; 4Department of Health and Sport Sciences, University of Louisville, Louisville, KY 40292, USA; coach_stam@rocketmail.com

**Keywords:** balance conditions, mood states, IMU sensors, POMS, classification modeling

## Abstract

Depressive mood states in healthy populations are prevalent but often under-reported. Biases exist in self-reporting of depression in otherwise healthy individuals. Gait and balance control can serve as objective markers for identifying those individuals, particularly in real-world settings. We utilized inertial measurement units (IMU) to measure gait and balance control. An exploratory, cross-sectional design was used to compare individuals who reported feeling depressed at the moment (*n* = 49) with those who did not (*n* = 84). The Quality Assessment Tool for Observational Cohort and Cross-sectional Studies was employed to ensure internal validity. We recruited 133 participants aged between 18–36 years from the university community. Various instruments were used to evaluate participants’ present depressive symptoms, sleep, gait, and balance. Gait and balance variables were used to detect depression, and participants were categorized into three groups: not depressed, mild depression, and moderate–high depression. Participant characteristics were analyzed using ANOVA and Kruskal–Wallis tests, and no significant differences were found in age, height, weight, BMI, and prior night’s sleep between the three groups. Classification models were utilized for depression detection. The most accurate model incorporated both gait and balance variables, yielding an accuracy rate of 84.91% for identifying individuals with moderate–high depression compared to non-depressed individuals.

## 1. Introduction

In 2017, the number of people worldwide who self-reported a diagnosis of depression exceeded 300 million [1]. However, this estimate does not account for individuals who do not meet the criteria for a clinical diagnosis of depression but experience mild to moderate levels of depression. Despite being considered healthy, such individuals may experience depressed moods that are aggravated by poor sleep, stress, and significant life events [2]. The subjective experiences of depressed moods can offer valuable insight into an individual’s mental health [3], and—although such moods alone are not sufficient to diagnose mental health disorders [4]—they often serve as a relative and attributable risk factor for the first-onset diagnosis of Major Depressive Disorder (MDD) [5]. Identification of individuals who report depressive moods and early intervention could help alleviate the burden of first-onset MDD [6].

The two most common methods for evaluating mood states are direct questioning by clinicians and the completion of self-reported mood state surveys, such as the Beck Depression Inventory [7], the state-trait depression survey [8], the Center for Epidemiological Studies Depression Scale (CES-D) [9], and the Profile of Mood States-Short Form (POMS-SF) [10]. However, the administration of self-reported surveys is plagued by the risk of social and cognitive biases [11]. These biases are often more pronounced in mental health assessments due to the stigma surrounding the issue [12]. Thus, it is essential to explore objective approaches to identify depressive mood states.

Biomarkers, such as high-sensitivity C-reactive protein (hs-CRP) and fibrinogen, have been linked to depressive moods in healthy individuals [13,14]. However, obtaining biological samples to identify biomarkers of depressive moods may exacerbate the stigma surrounding mental health and clinicians may not know when to order these tests. Therefore, it is crucial to investigate alternative objective markers for identifying individuals who report feelings of depression.

Recently, there has been growing interest in utilizing gait and machine learning to identify otherwise healthy individuals who report depressive mood states [15,16,17,18]. If implemented during wellness exams, gait-based measures of depression may remove the risk of biases, especially if the purpose was described as a health risk assessment. Although the accuracy rates of these studies [15,16,17,18] vary, they provide substantial evidence that gait may serve as a valid measure of assessing depressive mood states in healthy individuals. One limitation of these studies [15,16,17,18] is that they mainly measure depression as a mood experienced in the previous two weeks, which may not account for individuals experiencing recent life events that exacerbate depressive feelings at the moment. Another limitation is that these studies [15,16,17] employ two Xbox Kinect RGB-D cameras for video analysis, which may not be feasible in all settings.

Numerous studies have shown that depression significantly impairs balance control [19]. However, to the best of our knowledge, no research has investigated whether balance control can predict self-reported depressive moods. Although gait measures offer highly accurate ways of assessing depressive mood states in healthy, young adults [15,16,17,18], many of these studies require participants to walk around a 6 m track for about 2 min, which may not be feasible in real-world environments. Nevertheless, tests like the Modified Clinical Test of Sensory Interaction on Balance (mCTSIB) offer a means of assessing balance control in real-world settings, with conditions that can be easily replicated [20].

Psychological research on depression often focuses on individuals diagnosed with depression, resulting in a limited understanding of depressive mood states in healthy, young adults. However, it is important to recognize that even sub-threshold depressive mood states, which do not meet the diagnostic criteria for depression, can have significant negative impacts on young adults. These consequences include increased healthcare costs and reduced quality of life [21], diminished employee productivity [22], decreased academic productivity [23], and elevated mortality rates when compared to the general population without depressed mood [24]. Despite the high prevalence and substantial economic burden associated with depressive mood states, they tend to be under-recognized, under-diagnosed, and inadequately treated in clinical settings. In fact, less than one-third of adults with such mood states receive appropriate interventions and treatments [25]. Therefore, this study aimed to address this gap by examining a population of healthy young adults recruited from a university and its surrounding rural community.

This exploratory, hypothesis-generating study aimed to address three limitations in the literature, namely the reliance on individuals reporting depressive feelings over the past 2 weeks rather than in the moment, the use of data-intensive Xbox Kinect RGB-D cameras instead of inertial measurement unit (IMU) sensors, and the lack of research exploring the combined use of gait and balance control to predict depression. To this end, the study aimed to use IMU sensors to identify individuals who are currently experiencing depressive mood states, by utilizing a combination of gait and balance control measures.

## 2. Materials and Methods

### 2.1. Study Design

Given the ethical considerations involved in inducing feelings of depression in otherwise healthy individuals, the researchers employed a cross-sectional study design to compare individuals who reported feeling depressed at the moment with those who did not. The present study adhered to the principles outlined in the Declaration of Helsinki and obtained approval from the Institutional Review Board of Clarkson University (approval #18.39.1). Prior to participating in the study, all individuals involved provided written informed consent.

In addition, in order to ensure the internal validity of our study, we utilized the Quality Assessment Tool for Observational Cohort and Cross-sectional Studies, which was developed by the National Heart, Lung, and Blood Institute in 2013 [26]. This tool has been specifically designed to identify potential flaws in study methods and implementation for observational cohort and cross-sectional studies. By utilizing this tool, we were able to assess the quality of our study and ensure that it met the highest standards for internal validity. In Appendix A, we outline each criterion of the tool and provide an explanation of how it was applied to our study.

### 2.2. Participants

Participants aged between 18 and 36 years were recruited from the university community using various recruitment methods including word of mouth, flyers, campus-wide emails, and in-person announcements made in large classes with more than 30 students. Additionally, promotional flyers were strategically displayed at diverse establishments within the vicinity of the university’s location. Prospective participants expressing interest in the study were directed to complete an electronically administered survey, serving as an initial screening measure to ascertain their eligibility. Individuals with a diagnosis of neurological conditions, such as Parkinson’s disease, lower-extremity orthopedic surgeries, and/or injuries within the past 6 months, wounds on the plantar surface of their feet, uncorrectable visual impairments, acute or chronic mental illness, and inability to ambulate for 2 min without an assistive device or pain or discomfort, were excluded from participation. The full screening questionnaire can be found in the Appendix A section. Out of the 140 individuals who were screened, 133 met the eligibility criteria and were included in the study. Participants did not receive any form of compensation for their involvement in this study. Moreover, participants retained the autonomy to voluntarily withdraw from the study at any stage without incurring any penalties or adverse consequences.

### 2.3. Instruments

To evaluate participants’ present depressive symptoms, the depression component of the 30-item POMS-SF was utilized and the POMS-SF was administered immediately before the balance and gait assessment. Sleep data were collected through a questionnaire that inquired about various aspects of participants’ sleep immediately after completion of the balance and gait assessments. Gait was evaluated using OpalTM inertial movement unit sensors, while balance was assessed using the mCTSIB. Below is more information on the instruments used in the current study.

#### 2.3.1. Self-Reported Feelings of Depression

The current study utilized the depression component of the 30-item POMS-SF (21) to evaluate participants’ present depressive symptoms. The POMS-SF depression component required participants to self-report the intensity of their mood states on all 30 items using a 5-point Likert scale ranging from “not at all” (scored as 0) to “extremely” (scored as 4). Depressive symptoms were determined by summing scores across five questions (Sad + Unworthy + Discouraged + Lonely + Gloomy). The internal consistency of the current feelings of depression was high, as indicated by a Cronbach’s alpha of 0.897.

#### 2.3.2. Prior Night’s Sleep

Participants were asked to complete a written, open-ended questionnaire that inquired about various aspects of their sleep the night before the study. Specifically, they were asked to report the time they went to bed, the time they fell asleep, the time they woke up, the number of times they woke up in the middle of the night, and the duration of their awakenings. The duration of their sleep the prior night was calculated by subtracting the time of awakening from the time of falling asleep, minus the total duration of awakening. This survey has been previously used in multiple studies [27,28,29,30,31].

#### 2.3.3. Balance Assessment

To ensure a practical balance assessment in a real-world setting, without the need for expensive instruments such as force plates and the Neurocom Balance Master, we employed the modified Clinical Test of Sensory Interaction on Balance (mCTSIB), a balance measure that has a high correlation with the Sensory Organization Test [32] and can be easily completed using the OpalTM monitors. Participants underwent a 30 s test for four different conditions, with their feet spaced apart by the APDM footplate, and their hands on their hips during each condition. The four distinct conditions were: (1) eyes open with feet on the ground (EOG); (2) eyes closed with feet on the ground (ECG); (3) eyes open with feet on a foam surface (EOF) (Airex Balance Pad Foam Balance Board Stability Cushion, 50.5 cm W × 40.1 cm D × 6.0 cm H); and (4) eyes closed with feet on a foam surface (ECF). Balance assessment variables included measures assessing sway area, frequency domain characteristics, acceleration, jerk, mean velocity, and path length among others (a list of all variables is provided in the Appendix A and definitions of these variables may be found at www.apdm.com, accessed on 23 April 2023).

#### 2.3.4. Gait

Gait was evaluated in accordance with established methods for identifying individuals with depression based on walking gait. Similar to the protocol used by Zhao and colleagues [16], participants in this study were instructed to walk back and forth on a 6 m × 1 m track at a self-selected pace for a duration of two minutes. However, in contrast to Zhao et al., this study utilized OpalTM inertial measurement unit (IMU) sensors from the APDM Mobility LabTM (APDM Inc., Portland, OR, USA). These sensors are equipped with tri-axial accelerometers, gyroscopes, and magnetometers, and are synced wirelessly with a docking station to ensure sub-millisecond synchronization. The OpalTM monitors were attached to the lumbar region (5th lumbar vertebrae), sternum (body of sternum immediately superior to the xyphoid process), forehead middle of the frontal bone (approximately 2.5 cm above the nasal bone), right and left foot (on the metatarsals, directly superior to the metatarsophalangeal joint), and right and left wrist (immediately superior to the radioulnar joint) using VelcroTM straps [33,34]. Interested readers may consult our previously published studies for a detailed description of the gait assessment methodology [35,36]. Gait variables included but were not limited to anticipatory postural adjustment measures, cadence, gait speed, spatiotemporal variables of various gait cycle metrics, and planar angles of the neck, lumbar, leg, and foot segments in three dimensions among others (a list of all variables may be found in the Appendix A and definitions of these variables can be found at www.apdm.com, accessed on 23 April 2023)

### 2.4. Procedure

Following screening for inclusion/exclusion criteria, participants provided written informed consent and attended a single laboratory session lasting 75 min. Participants were instructed to abstain from consuming alcohol, caffeine, medications, and illicit drugs for 24 h prior to testing. Upon arrival at the laboratory, participants completed a battery of questionnaires on surveymonkey.com (San Mateo, CA, USA) regarding adherence to pre-testing instructions. Participants who did not comply with instructions were rescheduled.

Upon confirmation of eligibility on the day of the study, participants were randomly assigned a 5-digit identification number using the online tool randomizer.org and no identifiable information was collected from these participants. Participants were then equipped with APDM mobility monitors. Following the setup, the POMS-SF was administered to participants to assess their current emotional state. The survey was completed using a Hewlett Packard Pavilion 15.6″ Flagship Laptop (model #B018YIGHVK, Hewlett Packard, Palo Alto, CA, USA). The entire pre-testing procedure took approximately 10 min, with the POMS-SF being the final survey completed before participants underwent the mCTSIB (~2.5 min) and the two-minute walk test. The mCTSIB was conducted on one end of the walking track and immediately after completion, participants performed the two-minute walk test.

Following completion of the two-minute walk test, participant height was measured using a stadiometer (SECA model 220 Crothal Healthcare, Chino, CA, USA), and weight was measured using the Tanita Body Composition Analyzer TBF-410 (Tanita Corporation, Tokyo, Japan). Subsequently, participants were directed to complete a series of surveys aimed at providing researchers with Appendix A, including health-related lifestyle behaviors, which are outside the scope of this study.

### 2.5. Statistical Analyses

Pre-processing involved scoring mood surveys, extracting gait and balance data, and utilizing standard scaler and multiple imputation techniques. Feature importance was determined using random forest (RF), and various classifiers were used to classify individuals based on depression severity, with leave-one-out cross-validation and F1 scores used for assessment. Below, there is detailed information on the pre-processing steps, feature extraction, and classification methods used in this study. It is noteworthy that the outcomes assessors of the gait and balance tests were masked to the sleep and depression scores.

#### 2.5.1. Pre-Processing of Data

A comprehensive account of the pre-processing steps and calculation of gait variables is detailed in our previous publications [35,36]. In brief, the mood surveys obtained from surveymonkey.com were downloaded and scored using Microsoft Excel (Microsoft Inc., Redmond, WA, USA). The time series spatiotemporal gait and balance data were extracted from the APDM mobility monitor data files utilizing Python (version 3.7, Python, Software Foundation, Wilmington, DE, USA), which utilized data from the tri-axial accelerometer, gyroscope, and magnetometer. Following this, gait and balance characteristics were determined for all participants. The mood data was subsequently combined with the gait and balance data using Python (version 3.8.5). The data was initially fitted and transformed using Standard Scaler in Python (version 3.11) by utilizing the sklearn library. Missing data were then filled using multiple imputations using chained equations (MICE) [37]. Gait and balance data were visualized using the matplotlib library in Python, and individuals were grouped accordingly based on their depression score as assessed by the POMS-Depression scale. Specifically, individuals who reported a score of 0 on the POMS-Depression scale were classified as “Not Depressed,” while individuals who scored between 1 and 2 were classified as “Mild Depression,” and those who scored >3 were classified as “Moderate–High Depression.” The chi-square goodness of fit was used to determine sex differences, while analysis of covariance was used to determine differences in age, height, weight, and body mass index (BMI) between the three groups, utilizing the Pingouin library [38].

#### 2.5.2. Primary Analysis

Principal component analyses (PCA) using the sklearn library were conducted to create components for all gait variables, balance variables in each condition, balance variables in all conditions, and a combination of gait and balance variables in all conditions. Components with eigenvectors greater than 1.0 were considered significant. This study collected 310 features across all 5 conditions (gait, EOG, ECG, EOF, and ECF). To reduce the number of parameters for each analysis, feature importance was determined using random forest (RF) [39]. After feature extraction, various classifiers such as RF classifiers, K neighbors classifiers, support vector classifiers (SVC), decision tree classifiers, gradient boost classifiers, ada boost classifiers, Gaussian naïve Bayesian classifiers, linear discriminant analysis classifiers, quadradic discriminant analysis classifiers, and stochastic gradient descent classifiers were used to classify individuals and identify best algorithms for this exploratory work. Given the small sample size, classifiers were run to identify Not Depressed vs. Mild Depression, Not Depressed vs. Moderate–High Depression, and Mild Depression vs. Moderate–High Depression, using a combination of gait alone and all balance conditions separately, all balance conditions together, gait and balance together, further PCAs of gait, each balance condition separately, all balance conditions, and balance and gait conditions combined, were used as predictors. All variables, as well as top variables, were used for classifiers. Due to the small sample size, a leave-one-out (LOO) cross-validation method [40] was employed, and F1 scores were used to assess sensitivity [41] given the imbalanced nature of the data.

#### 2.5.3. Post Hoc Analysis

Depending on the distribution of data, post hoc analysis of variance (ANOVAs) or Kruskal–Wallis tests were used to assess differences in gait and balance variables using the Pingouin library [38]. Pairwise *t*-tests or Mann–Whitney U tests, with Bonferroni adjustments, were used for post hoc analyses where necessary.

## 3. Results

The results of all classifier analyses, feature importance, and ANOVAs can be found in the Appendix A. In the present study, only the predictors and significant findings for the ANOVAs and Kruskal–Wallis tests have been presented in the results.

### 3.1. Participant Characteristics

Among the 133 individuals eligible for this study, 84 individuals (32 males, 52 females) reported no feelings of depression at the time of assessment, while 27 individuals (10 males, 17 females) reported scores of 1 or 2 on the POMS-Depression scale and were classified as “Mild Depression”. The remaining 22 individuals (8 males, 14 females) reported scores > 2 on the POMS-Depression scale and were classified as “Moderate–High Depression.” Chi-square analysis revealed no significant differences in the number of males and females across the three groups. ANOVA findings showed that there were no significant differences in age, height, weight, BMI, and prior night’s sleep between the three groups (refer to Table 1). Thus, these variables were not included as predictors in the subsequent classification analyses.

### 3.2. Classification Models for Depression Detection Using Gait and Balance Variables

The PCA analyses yielded 37 components for gait, 5 components for the eyes open and feet on the ground condition, and components for the eyes closed and feet on the ground condition, 6 components for the eyes open and feet on the foam surface condition, and the eyes closed and feet on the foam surface condition, 19 components for all balance conditions, and 48 components for all variables. Detailed results including the weights of each variable in the components can be found in the Appendix A.

#### 3.2.1. Not Depressed vs. Mild Depression

The Appendix A provides a detailed report on feature importance for all variables. Among the classification models, the most accurate model for identifying individuals with mild depression compared to non-depressed individuals incorporated both gait and balance variables through principal component analyses. The random forest classifier exhibited an accuracy rate of 77.68% and an F1 score of 0.78. Other models demonstrated comparable accuracy rates ranging from 75% to 77.68% with F1 scores ranging from 0.75 to 0.77 (see Table 2). Post hoc findings provide evidence that compared with not depressed individuals, those with mild depression walked with greater lateral acceleration anticipatory postural adjustments and reduced leg elevation at mid-swing. In the ECF balance condition, those with mild depression had reduced 95% ellipse area radius of acceleration in the sagittal direction and reduced acceleration sagittal path length, root mean square (RMS) sway, and RMS sway in the sagittal plane, as well as reduced 95% ellipse area sagittal sway angle, RMS sway angle, and RMS sway angle in the sagittal plane compared with not depressed (Table 3).

#### 3.2.2. Not Depressed vs. Moderate to High Depression

All the top-performing models in the present study exhibited accuracy rates ≥79%, with F1 scores ranging from 0.79 to 0.84. Notably, the most accurate model was found to incorporate both the gait and all balance conditions, yielding an accuracy rate of 84.91% and an F1 score of 0.84 (Table 2). Post hoc analyses suggest that individuals who are depressed have reduced leg elevation at mid-swing and less variance between legs in their mid-swing elevation (Table 3).

#### 3.2.3. Mild Depression vs. Moderate to High Depression

The accuracy rates of the top models varied from 58% to 80%, with corresponding F1 scores ranging from 0.46 to 0.82. Notably, the model that performed the best incorporated both gait and balance conditions, utilizing the top values obtained from principal component analyses. This model achieved an accuracy rate of 80% and an F1 score of 0.82 (see Table 2). Findings of the ANOVAs suggest that individuals who reported moderate to high depression had significantly slower forward anticipatory postural adjustment while they were walking compared to those who reported mild feelings of depression. In the ECF condition, individuals who reported mild feelings of depression had reduced acceleration in the sagittal plane, RMS, and RMS in the sagittal plane and reduced sway angles in the sagittal plane, RMS sway, and RMS sway in the sagittal plane (Table 3).

## 4. Discussion

The objective of this exploratory, hypothesis-generating study was to identify individuals reporting mild or moderate to high depression using walking gait and balance conditions. The study found that the most accurate way to identify between the three classes is to use the two-minute walking gait and all four balance conditions of the mCTSIB. However, when identifying mild depression, the accuracy rates and F1 scores of models with all conditions and each individual condition were not significantly different. Similarly, most models for moderate to high depression had similar accuracy rates. The model that accounted for both gait and all balance conditions had the highest accuracy rate and F1 score. In conclusion, depressive mood states can be identified in young, healthy individuals using a walking gait and the mCTSIB.

Our findings indicate that our models were more accurate than those of Zhao et al. [16], but less accurate than those of Lu et al. [15], Yang et al. [18], and Fang et al. [17]. We note that our methodology for detecting feelings of depression using gait was the same as that of Zhao et al. [16], but our best models using gait only were more accurate than theirs. While we cannot speculate as to why our models had better predictive accuracies, some potential explanations may be that Zhao et al. [16] selected approximately 2 s of data and used a 10 fold cross-validation technique, which may have decreased their accuracy rates. In contrast, we chose to use the entire 2 min of gait and a leave-one-out cross-validation method, as we did not feel that our sample size was large enough to warrant the use of k-fold cross-validation. Our methodologies differed from those of the other three studies [15,17,18], which may explain the differences in accuracy rates. Specifically, Lu et al. [15] utilized a 10 m walk, our study only used a 6 m walk. Lu et al. [15] utilized a 10 m walk and extracted specific joint mechanical energy in walking gait, while we used kinematic features that could be extracted from IMU sensors and only used a 6 m walk. Another difference between our study and previous studies [15,16,17,18] is that we attempted to identify three classes, not depressed, mild depression, and moderate to high depression, whereas others used a dichotomous decision measure for clinical depression. Thus, an advantage of our study is that it can differentiate between levels of depression including levels that do not meet criteria for diagnosis of clinical depression.

The present study compared its classification methodology with that of Fang et al. [17], who also employed various classifiers to detect individuals with depression. However, our study differed from their work in that they focused on feelings of depression over the past two weeks, while we aimed to identify current depressive states. Furthermore, the study by Fang et al. [17], had a larger sample size of over 3600 participants, which may have contributed to higher accuracy rates. Of note, Yang et al. [18] reported the most accurate results among studies identified that utilized gait to predict depression in healthy young adults [15,16,17]. It is worth mentioning that the use of deep learning in their analyses may have accounted for the improved accuracy rates.

Interestingly, in our post hoc analyses, significant differences between the three groups were only present in the balance condition where participants were maximally challenged, while our machine learning models did not differ much between each of the balance conditions. However, the results of the post hoc analysis provide significant insight into the relationship between feelings of depression and balance, as our findings suggest that the relationship between depression and balance is non-linear. For example, individuals who report mild feelings of depression have more constrained balance, which could indicate increased tension and/or stress, while individuals who are not stressed are less constrained and more “carefree”. Interestingly, individuals who reported moderate to high depression were also not as constrained as those who reported mild depression, which may reflect a sense of having “given up” rather than a carefree outlook, given that their balance parameters did not differ from those who reported no depression. These findings are contrary to the literature that has identified balance differences between individuals with MDD compared to health controls [19]. However, our study was different in that the adults in our study did not qualify for a diagnosis of MDD. This non-linear relationship between human movement and depression is further supported by the differences in anticipatory postural control in gait, where individuals who report mild depressive moods have slower forward postural adjustments during initiation of gait, while individuals who were not depressed or report moderate-to-high depression have significantly faster forward anticipatory postural adjustments during gait initiation. Further, in line with the previous literature [42,43], not depressed individuals had less constrained walking gait, which may allow them to respond more efficiently to perturbations [43].

### 4.1. Practical Implications

The results of this exploratory study may have practical implications for researchers and practitioners. Researchers interested in identifying transient feelings of depression in non-clinical healthy populations may need to account for the fact that, like in our population, differences in balance and gait were non-linear. Future researchers should try to minimize the number of sensors (i.e., using smartwatches with IMU sensors) to see if it is possible to identify feelings of depression in a more natural setting. The further development of these algorithms may be beneficial for providing insights to populations with autism spectrum disorder (ASD) who may have a hard time recognizing affect [44]. Depression is also concurrent with many chronic diseases [45] and further development of these algorithms may guide clinicians who are treating other chronic diseases to also address feelings of depression in populations who may not meet the clinical criteria for a diagnosis of MDD.

### 4.2. Limitations

The present exploratory study has some limitations that should be noted. Firstly, the study had a cross-sectional design with a sample size of 133 total participants, and only 27 participants with mild depression and 22 with moderate–high depression, which may limit the ability to establish causality and evaluate external validity. A longitudinal study that tracks participants’ moods over time, similar to the study by Sukei et al. [46], may have been more appropriate. However, as we were interested in identifying feelings of depression at the moment using IMU sensors, a longitudinal design may have been impractical. Secondly, the study was conducted with a limited population of healthy young adults, and thus, the generalizability of the findings within and beyond this population is uncertain. Future research should consider examining larger and more diverse populations. Furthermore, mood was assessed through self-report measures which may have introduced inherent biases into the results [47]. Another potential limitation is the use of lab-based gait and balance measurements which may not be reflective of natural settings [48]. Additionally, the study did not differentiate between subtypes of depression [49], which may have distinct gait features. The Quality Assessment Tool for Observational Cohort and Cross-sectional Studies was used to assess the quality of our study, which met the highest standards for internal validity. While our study did not fully meet all the criteria outlined by this tool, we provided justifiable explanations based on our study design’s specificities, accounting for potential limitations and ensuring the methodology’s appropriateness and relevance to the research question. Thus, our study maintained a high level of internal validity and provided reliable results despite not satisfying all criteria. Another potential limitation of this study is that we were unable objectively verify the veracity of the self-reported medical information provided by the participants. Furthermore, although our study did not comprehensively examine all possible biomechanical factors that may influence gait patterns, we acknowledge the potential contributions of such factors. Future research with a more extensive assessment of biomechanical parameters is warranted to gain a more comprehensive understanding of the relationship between gait and depression. Lastly, the use of 7 IMU sensors may not be generalizable to other ways of collecting data in real-world settings. Future development of this work could lead to easier testing of depression mood states associated with gait abnormalities using a single wearable device that can be implemented quickly during wellness exams, thus addressing the limitations of the current depression assessment techniques.

## 5. Conclusions

The aim of this exploratory study was to utilize IMU sensors to identify current feelings of depression in a cohort of young, healthy adults by measuring gait and balance. A validated mood state scale was employed to classify individuals into three categories: not depressed, mild depression, and moderate to high depression. Through the implementation of principal component analysis (PCA) and various classification models, we were able to accurately classify individuals into these three categories, achieving accuracy rates ranging from 77% to 81%. Future researchers should generate hypotheses and expand on our work by utilizing larger and more diverse samples and examining whether single IMU sensors can detect changes in mood states over time in these populations.

## Figures and Tables

**Table 1 sensors-23-06624-t001:** Participant characteristics.

Characteristic	Not Depressed	Mild Depression	Moderate–High Depression	*p*-Value
Sex (M:F)	32:52	10:17	8:14	0.557
Age (years)	26.15 ± 8.47	25.37 ± 6.78	24.95 ± 7.50	0.783
Height (cm)	172.82 ± 8.69	173.85 ± 7.47	172.95 ± 10.48	0.873
Weight (kg)	74.49 ± 14.92	74.56 ± 13.96	73.44 ± 18.78	0.957
BMI (kg/m^2^)	24.85 ± 4.67	24.50 ± 3.56	24.43 ± 5.58	0.897
Prior night’s sleep (hours)	7.49 ± 1.70	7.54 ± 1.34	7.82 ± 1.72	0.704

**Table 2 sensors-23-06624-t002:** Top classifier results only.

Variable	Not Depressed vs. Mild Depression			Not Depressed vs. Moderate–High Depression			Mild Depression vs. Moderate–High Depression		
	Model Name	Mean Accuracy	F-1 Scores	Model Name	Mean Accuracy	F-1 Scores	Model Name	Mean Accuracy	F-1 Scores
Gait									
All Variables	SVC	75.00%	0.75	Random Forest/SVC	79.25%	0.79	QDC	72.00%	0.72
Top Variables	Random Forest	77.68%	0.77	Gaussian NB/LDA	81.13%	0.81	Gaussian NB	70.00%	0.70
All Components	SVC/QDA	75.00%	0.75	KNeighbor	80.19%	0.80	Random Forest	58.00%	0.46
Top Components	Random Forest	77.68%	0.76	KNeighbor	80.19%	0.80	KNeighbor	64.00%	0.64
Balance—Eyes Open, Feet on Ground									
All Variables	SVC/QDA	75.00%	0.75	GaussianNB	80.19%	0.80	LDA	64.00%	0.64
Top Variables	KNeighbor	75.89%	0.76	Random Forest	81.13%	0.80	Gradient Boosting	60.00%	0.60
All Components	KNeighbor	75.89%	0.76	SVC/LDA	79.25%	0.79	Gaussian NB	60.00%	0.60
Top Components	KNeighbor	75.89%	0.76	Random Forest	81.13%	0.79	Gaussian NB	60.00%	0.60
Balance—Eyes Closed, Feet on Ground									
All Variables	SVC/QDA	75.00%	0.75	SVC/QDA	79.25%	0.79	Gradient Boosting	68.00	0.66
Top Variables	SVC	75.00%	0.75	KNeighbor	80.19%	0.80	Decision Tree	76.00%	0.72
All Components	SVC/LDA	75.00%	0.75	Random Forest	80.19%	0.78	QDA	60.00%	0.60
Top Components	SVC/LDA	75.00%	0.75	KNeighbor/SVC	79.25%	0.79	QDA	60.00%	0.60
Balance—Eyes Open, Feet on Foam Surface									
All Variables	SVC/QDA	75.00%	0.75	SVC/QDA	79.25%	0.79	KNeighbor	58.00%	0.58
Top Variables	SVC/LDA	75.00%	0.75	SVC	79.25%	0.79	Gaussian NB	62.00%	0.62
All Components	SVC	75.00%	0.75	SVC/GaussianNB	79.25%	0.79	QDA	76.00%	0.76
Top Components	SVC	75.00%	0.75	SVC	79.25%	0.79	QDA	66.00%	0.66
Balance—Eyes Closed, Feet on Foam Surface									
All Variables	SVC/QDA	75.00%	0.75	SVC/QDA	79.25%	0.79	QDA	64.00%	0.64
Top Variables	KNeighbor	75.89%	0.76	SVC	79.25%	0.79	SVC	64.00%	0.64
All Components	SVC	75.00%	0.75	SVC	79.25%	0.79	Decision Tree	72.00%	0.68
Top Components	SVC	75.00	0.75	SVC/Gaussian NB	79.25%	0.79	Decision Tree	74.00%	0.64
All Balance Conditions									
All Variables	SVC	75.00%	0.75	Random Forest	79.25%	0.80	Gradient Boosting	66.00%	0.62
Top Variables	KNeighbor/SVC	75.00%	0.75	Random Forest/Gradient Boosting	79.25%	0.80	AdaBoost	68.00%	0.68
All Components	SVC	75.00%	0.75	KNeighbor/SVC	79.25%	0.79	QDA	60.00%	0.60
Top Components	SVC/Gradient Boost	75.00%	0.75	Random Forest/KNeighbor/SVC	79.25%	0.79	KNeighbor	72.00%	0.72
All Balance Conditions and Gait									
All Variables	SVC	75.00%	0.75	Random Forest	79.25%	0.80	AdaBoost	60.00%	0.60
Top Variables	AdaBoost	77.68%	0.78	Gradient Boosting	84.91%	0.84	SVC	72.00%	0.72
All Components	Random Forest/SVC/QDA	75.00%	0.75	Random Forest/SVC/QDA	79.25%	0.79	Decision Tree	70.00%	0.78
Top Components	Random Forest	77.68%	0.78	LDA	83.02%	0.83	Random Forest	80.00%	0.82

**Table 3 sensors-23-06624-t003:** Significant findings post hoc ANOVAs—mean (SD).

Measure	No Depression	Mild Depression	Moderate–High Depression	Post Hoc
Gait Features				
Anticipatory Postural Adjustment Forward Peak (m/s^2^)	0.33 (0.17)	0.25 (0.17)	0.36 (0.21)	Moderate–High > Mild
Anticipatory Postural Adjustment Lateral Peak (m/s^2^)	0.37 (0.21)	0.28 (0.18)	0.42 (0.19)	No, Moderate–High > Mild
Mid-swing Leg Elevation Variance within Leg	0.38 (0.11)	0.44 (0.15)	0.44 (0.16)	No < Mild, Moderate–High
Variance Between Legs Mid-swing Elevation	19.54 (15.65)	16.76 (15.59)	10.80 (9.71)	No > Moderate–High
Eyes Closed Feet on Foam Surface				
Acceleration 95% Ellipse Y-Axis Radius (m/s^2^)	0.26 (0.10)	0.21 (0.06)	0.25 (0.06)	No > Mild
Acceleration Path Length in Sagittal Plane (m/s^2^)	8.91 (3.26)	7.08 (2.08)	8.62 (3.19)	Mild < No, Moderate–High
Acceleration Root Mean Square Sway (m/s^2^)	0.12 (0.04)	0.10 (0.03)	0.12 (0.03)	Mild < No, Moderate–High
Acceleration Root Mean Square Sway in Sagittal Plane (m/s^2^)	0.10 (0.04)	0.09 (0.03)	0.10 (0.03)	Mild < No, Moderate–High
Sway Angle 95% Ellipse Y-Axis Radius (°)	1.54 (0.57)	1.25 (0.36)	1.46 (0.36)	Mild < No, Moderate–High
Root Mean Square Sway Angle (°)	0.71 (0.25)	0.59 (0.16)	0.69 (0.16)	Mild < No, Moderate–High
Root Mean Square Sway Angle in Sagittal Plane (°)	0.62 (0.24)	0.51 (0.15)	0.59 (0.15)	Mild < No, Moderate–High

## Data Availability

The data presented in this study are available on request from the corresponding author. The data are not publicly available due to IRB restrictions.

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
