# Peer review of "Identifying Current Feelings of Mild and Moderate to High Depression in Young, Healthy Individuals Using Gait and Balance: An Exploratory Study"

_sensors, 2023, doi:10.3390/s23146624_

Round 1
Reviewer 1 Report
Depression along with other mental health challenges are on the rise within several countries and this brings about urgency within the mental health community on a world stage as to how to address mental health in an effective and efficient manner. This study comes at a critical time and addresses the growing problem as early diagnosis is often key in an effective therapeutic intervention. With trends seen in the advances in technology as it relates to diagnostic tools, this too emphasizes the importance of this study, and thus, the authors are commended for developing a manuscript that is relevant and applicable within the mental health community. The introduction does a solid job of laying down a foundation for the background and reasoning for the study. The only addition that should be addressed within the introduction is the importance of why this study focused on young adults as stated in the title. Justification is needed within the introduction and the objectives need to clearly state that the focus of this study is young adults. With the high rates of suicide and substance abuse within the young adult population, this makes the study extremely critical as early detection of depression may redirect a young adult from leading down the road to suicide or substance abuse.
As for the methods, this section needs to begin with a clear statement concerning research protocol approval by a University-based ethics review community concerning research using vulnerable human subjects. Human research subjects suffering from mental health disorders, even classified as mild, would fall under the classification of vulnerable subjects, and thus, requiring clear statements that ethical review was completed and protocol approval was accomplished. Clear statements concerning consent of not only research participation, but also consent to allow the use of medical history, which is sensitive information and can be considered identifiable, needs to be given. How was confidentiality ensured? Were study participants compensated in any form? Were all aspects of the study voluntary and could they refuse participation at any point within the study? This information needs to be clearly stated within the methods.
The authors are commended for their understanding of the limitations associated with not having a control for the research study. This is not uncommon in mental health studies, however, the utilization of the Quality Assessment Tool for Observational Cohort and Cross-Sectional Studies demonstrated an extra step in ensuring the validity of the work presented. Nevertheless, the table, while interesting, could be included as supplemental materials, rather than found within the main text. Leave in the text information about what was done and the overall results associated with this assessment method, but just refer within the text that the table concerning specifics of the assessment can be found within supplemental materials. For the participants, how was the health information verified other than the information given by the participants themselves? Were health records supplied by their physicians for further verification or was it a self-reporting survey? If self-reporting, how was reliability verified? Furthermore, if a self-reporting survey, was it a written survey or an oral interview and what specifically were the questions? Questions should be included within supplemental materials and clear, objective inclusion/exclusion criteria needs to be included within the text. How was soundness of limbs and gait objectively verified at the time of the study to ensure no unforeseen injury and/or strain that could impact measurements for that day? Was medical assessment done by a physician and/or nurse? For the questionnaire on sleep was that done at a separate time from the questions related to the health utilized for inclusion/exclusion criteria? Was it written or oral questions? Again, these questions need to be given within this paper and options of what type of answers could be utilized, ie. closed or open ended, and the length of the questionnaires need to be available. While length and type of questionnaire for the depression questions were clearly indicated, as mentioned previously, limited information on the sleep and medical information questionnaires is given. Along with the length and type of questions, how were these questions developed, ie. by the authors, and how were the questions validated as to their application for this study? For balance assessment variables and gait variables give a clear, objective definition of each of these variables. Velocity was allowed to be selected individually by participants, however, gait variables are dependent on velocity, and thus, justification for not controlling for velocity should be given and explanation should be included on how velocity variability was accounted for in assessment of the data.
As for the results, the sample size is concerning as 84 out of the 133 did not have any depression and only 22 demonstrated moderate to high depression with only 8 males. This, when breaking down to the actual numbers per subgroups assessed, fits more the definition of what a pilot study would entail when you think about the fact that these findings are given as if 22 people represent the entire population of young adults with depression. The authors do mention within the discussion comparisons to another study with 3600 participants indicating the accuracy of that study over the current study due to a larger sample size. Thus, within limitations within the discussion section the small sample population should be further explored. The authors need to also clearly indicate in the abstract the true breakdown of the numbers utilized within the sample population of the study. Within this same section of the discussion concerning differences with other studies, the authors bring up the uniqueness of their study for tracking current depressive states. Obviously, this is a unique factor to the study, and thus, why is this not a part of the title? Adjust title accordingly. As for further development within the discussion section, the limitations of the study, while authors hit on some main points within this section, should be expanded and should include references supporting these conclusions concerning limitations. Finally, the conclusions should be more streamlined and focused more on the results than that of hindering factors. Leave the limitations within the section labeled limitations and what statement is made concerning future directions should stay targeted on how this research laid a foundation for the next step rather than where it fell short.
Author Response
Dear Reviewer,
We would like to thank the reviewer for their time and excellent feedback. We feel that based on the reviewer’s feedback our manuscript is significantly improved.
Reviewer 1
Depression along with other mental health challenges are on the rise within several countries and this brings about urgency within the mental health community on a world stage as to how to address mental health in an effective and efficient manner. This study comes at a critical time and addresses the growing problem as early diagnosis is often key in an effective therapeutic intervention. With trends seen in the advances in technology as it relates to diagnostic tools, this too emphasizes the importance of this study, and thus, the authors are commended for developing a manuscript that is relevant and applicable within the mental health community.
We thank the reviewer for their kind words.
The introduction does a solid job of laying down a foundation for the background and reasoning for the study.
Thank you.
The only addition that should be addressed within the introduction is the importance of why this study focused on young adults as stated in the title. Justification is needed within the introduction and the objectives need to clearly state that the focus of this study is young adults. With the high rates of suicide and substance abuse within the young adult population, this makes the study extremely critical as early detection of depression may redirect a young adult from leading down the road to suicide or substance abuse.
We sincerely appreciate the insightful feedback provided by the reviewer, which allows us to elaborate on our rationale for selecting healthy, young adults as the focus of this study. While acknowledging the reviewer's valid points regarding the high rates of suicide and substance abuse in this demographic, we would like to emphasize that our decision was motivated by the under-explored nature of depressive mood states among young, healthy individuals. Despite the significant burden imposed by these mood states, they often receive insufficient attention in research, even though they carry elevated risks for future mortality. Moreover, we would like to take this opportunity to clarify that our study encompassed not only a university population but also a broader population of young adults residing in a small rural town, including those who were not enrolled as college students. To address these points, we have updated the objectives to explicitly highlight the focus on healthy, young adults.
As for the methods, this section needs to begin with a clear statement concerning research protocol approval by a University-based ethics review community concerning research using vulnerable human subjects. Human research subjects suffering from mental health disorders, even classified as mild, would fall under the classification of vulnerable subjects, and thus, requiring clear statements that ethical review was completed and protocol approval was accomplished. Clear statements concerning consent of not only research participation, but also consent to allow the use of medical history, which is sensitive information and can be considered identifiable, needs to be given.
We sincerely appreciate the insightful feedback provided by the reviewer, which allows us to elaborate on our rationale for selecting healthy, young adults as the focus of this study. While acknowledging the reviewer's valid points regarding the high rates of suicide and substance abuse in this demographic, we would like to emphasize that our decision was motivated by the under-explored nature of depressive mood states among young, healthy individuals. Despite the significant burden imposed by these mood states, they often receive insufficient attention in research, even though they carry elevated risks for future mortality. Moreover, we would like to take this opportunity to clarify that our study encompassed not only a university population but also a broader population of young adults residing in a small rural town, including those who were not enrolled as college students. To address these points, we have updated the objectives to explicitly highlight the focus on healthy, young adults.
How was confidentiality ensured?
This information has been provided in the procedure section where we mention that participants were given a random 5-digit ID and no identifiable information was collected from these participants.
Were study participants compensated in any form?
We have now added a statement stating that participants were not compensated for their participation in the study.
Were all aspects of the study voluntary and could they refuse participation at any point within the study? This information needs to be clearly stated within the methods.
We have now clearly stated this information in the Participants section of the methodology.
The authors are commended for their understanding of the limitations associated with not having a control for the research study. This is not uncommon in mental health studies, however, the utilization of the Quality Assessment Tool for Observational Cohort and Cross-Sectional Studies demonstrated an extra step in ensuring the validity of the work presented. Nevertheless, the table, while interesting, could be included as supplemental materials, rather than found within the main text. Leave in the text information about what was done and the overall results associated with this assessment method, but just refer within the text that the table concerning specifics of the assessment can be found within supplemental materials.
We appreciate this feedback and have now moved the Quality Assessment Tool to the supplemental materials section.
For the participants, how was the health information verified other than the information given by the participants themselves?
Health information was not verified. If participants answered the questions “Yes” to any of the questions that asked them if they had certain health condition, they were not allowed to participate in the study.
Were health records supplied by their physicians for further verification or was it a self-reporting survey? If self-reporting, how was reliability verified?
The participants provided self-reported information. Due to IRB constraints, we were not allowed to verify whether the participants who stated that they did not have any of the health conditions we asked about. We have now addressed this in our limitations.
Furthermore, if a self-reporting survey, was it a written survey or an oral interview and what specifically were the questions? Questions should be included within supplemental materials and clear, objective inclusion/exclusion criteria needs to be included within the text.
Participants were asked to complete a screening survey prior to being invited to lab. We have provided the inclusion/exclusion criteria in the text in the Participants section. Further we have now included the inclusion/exclusion screening survey in the supplemental section.
How was soundness of limbs and gait objectively verified at the time of the study to ensure no unforeseen injury and/or strain that could impact measurements for that day?
Due to IRB constraints, we were unable to examine objectively identify unforeseen injuries and/or strains that could impact their gait measurements. We were only allowed to confirm using a self-reported survey with the participants whether they had a change in their health status since completing the original screening questionnaire. If an obvious gait or balance issue was observed during testing, the subject would have been excluded from the study. We have now addressed this in our limitations section.
Was medical assessment done by a physician and/or nurse?
A medical assessment was not completed by a physician and/or nurse since this study did not include any medical information.
For the questionnaire on sleep was that done at a separate time from the questions related to the health utilized for inclusion/exclusion criteria? Was it written or oral questions? Again, these questions need to be given within this paper and options of what type of answers could be utilized, ie. closed or open ended, and the length of the questionnaires need to be available. While length and type of questionnaire for the depression questions were clearly indicated, as mentioned previously, limited information on the sleep and medical information questionnaires is given. Along with the length and type of questions, how were these questions developed, ie. by the authors, and how were the questions validated as to their application for this study?
We have now clarified that the sleep questionnaire was completed on the day of the study and not during the inclusion/exclusion criteria. We have provided information that this was a written questionnaire. This questionnaire is open-ended questionnaire. The authors used a survey that has previously been used in multiple studies.
For balance assessment variables and gait variables give a clear, objective definition of each of these variables.
We appreciate the reviewer’s feedback; we have now provided the reader with information that the lay definition of each of the variables used in our study may be found at www.apdm.com. If the reviewer prefers, we can also create a supplementary table to provide the reader with definitions of these variables.
Velocity was allowed to be selected individually by participants, however, gait variables are dependent on velocity, and thus, justification for not controlling for velocity should be given and explanation should be included on how velocity variability was accounted for in assessment of the data.
We had the subjects walk at a self-selected speed rather than a standardized walking speed because depressive mood may influence gait speed, and thus the speed dependent features of gait. Average gait speed is one of the variables evaluated by the analysis and so between subjects variability in gait speed was accounted for.
As for the results, the sample size is concerning as 84 out of the 133 did not have any depression and only 22 demonstrated moderate to high depression with only 8 males. This, when breaking down to the actual numbers per subgroups assessed, fits more the definition of what a pilot study would entail when you think about the fact that these findings are given as if 22 people represent the entire population of young adults with depression.
Could the reviewer please clarify this statement or provide an example and justification for a more ideal sample? The authors do not suggest that the 22 adults with moderate to high depression represent the entire population of young adults with depression. A balanced sample across all three groups was not expected, yet the proportion with depressive symptoms (approx. 36%) exceeds the rate of depression in young adults as reported by Columbia University.
The authors do mention within the discussion comparisons to another study with 3600 participants indicating the accuracy of that study over the current study due to a larger sample size. Thus, within limitations within the discussion section the small sample population should be further explored.
We have now written about the small sample size as a potential limitation.
The authors need to also clearly indicate in the abstract the true breakdown of the numbers utilized within the sample population of the study.
The abstract has been revised to indicate the number of those who did and did not feel depressed at the moment.
Within this same section of the discussion concerning differences with other studies, the authors bring up the uniqueness of their study for tracking current depressive states. Obviously, this is a unique factor to the study, and thus, why is this not a part of the title? Adjust title accordingly.
We appreciate the reviewer understanding this as a uniqueness of this study. We have adjusted the title accordingly.
As for further development within the discussion section, the limitations of the study, while authors hit on some main points within this section, should be expanded and should include references supporting these conclusions concerning limitations.
We appreciate the reviewer’s comments and have now added references where we felt one could be provided. However, we welcome the reviewer to specify any information that still requires a citation, if remaining.
Finally, the conclusions should be more streamlined and focused more on the results than that of hindering factors. Leave the limitations within the section labeled limitations and what statement is made concerning future directions should stay targeted on how this research laid a foundation for the next step rather than where it fell short.
We appreciate this feedback from the reviewer and have now moved the limitations of the study to the limitations section. We have now focused our conclusion on the future direction that this research has laid the foundation for.

Reviewer 2 Report
In this study, the authors attempted to identify patients with depression using gait and balance data and machine learning techniques. They collected gait data using IMU from 133 participants ranked from no depression to mild and high depression.
Although the authors did a proper job of collecting the gait and balance data; however, the paper is not scientifically significant. As the main theme of the study is gait deviation, it requires firm hypotheses regarding the biomechanics of walking. Gait deviation is coming from motor control impairment (e.g. patient with cerebral palsy) or musculoskeletal deficits (e.g. leg amputee). The gait itself is quite robust meaning without strong intervention there would be no significant changes in its pattern. Using the data mining models in a black box manner without any biomechanical supervision provides no meaningful results. There are no clear hypotheses or biomechanics discussions in the paper. Personally, I don’t see any reason that these participants have musculoskeletal incompetence (e.g. high usage of specific drugs that may affect motor control or cause muscle weakness), thus I can’t relate the outputs to the depression level.
No wonder you get some results but that doesn’t necessarily is related to categorizing the walking based on depression level. As soon as you increase the degree of freedom of the model (e.g. number of inputs) your output converges to some extent. The balance test is not robust as well, as the subjects were pushed to the limits. Your averaged accuracy rates ranging from 77% to 81% don’t really imply the success of the robust classifying either. There could be several biomechanical reasons which have been neglected here.
Overall this paper might’ve been an interesting in-house study but as a scientific study unfortunately I don’t see that much value in that, aand unfortunately, I have to suggest rejection.
The entire script is well written and understandable.
Author Response
Dear Reviewer,
We would like to thank the reviewer for their time and excellent feedback. We feel that based on the reviewer’s feedback our manuscript is significantly improved.
Reviewer 1
Reviewer 2
In this study, the authors attempted to identify patients with depression using gait and balance data and machine learning techniques. They collected gait data using IMU from 133 participants ranked from no depression to mild and high depression.
Although the authors did a proper job of collecting the gait and balance data; however, the paper is not scientifically significant. As the main theme of the study is gait deviation, it requires firm hypotheses regarding the biomechanics of walking. Gait deviation is coming from motor control impairment (e.g. patient with cerebral palsy) or musculoskeletal deficits (e.g. leg amputee). The gait itself is quite robust meaning without strong intervention there would be no significant changes in its pattern. Using the data mining models in a black box manner without any biomechanical supervision provides no meaningful results. There are no clear hypotheses or biomechanics discussions in the paper. Personally, I don’t see any reason that these participants have musculoskeletal incompetence (e.g. high usage of specific drugs that may affect motor control or cause muscle weakness), thus I can’t relate the outputs to the depression level.
We appreciate the reviewer's evaluation of our study; however, we respectfully disagree with their assessment of the scientific significance. While the main focus of our study is gait deviation, we would like to emphasize that gait abnormalities can arise from various factors, including both motor control impairments (e.g., cerebral palsy) and musculoskeletal deficits (e.g., leg amputations). It is important to note that gait itself is not solely dependent on these factors, but is also influenced by higher cortical processes involving motor planning, coordination, and cognitive aspects.
Contrary to the reviewer's suggestion, there is substantial evidence in the existing literature supporting the influence of mood on gait patterns. Numerous studies have demonstrated the impact of various mood states, including depression, anxiety, fatigue, and anger, on gait parameters (please see list of partial studies below). We have referenced a meta-analysis and other relevant studies in the introduction, which provide substantial evidence for the influence of moods on gait biomechanics. These findings indicate that gait deviations can be linked to cognitive loads, changes in mood, and other psychological factors.
In our study, we purposefully excluded individuals with neurological or musculoskeletal issues to ensure that the observed gait deviations were primarily influenced by mood factors. By addressing the gaps identified in the literature, as outlined in the final paragraph of the introduction, our study aims to contribute to the development of unbiased methods for assessing depression, leading to improved detection and intervention strategies. We are more than willing to provide the reviewer with additional studies that support the influence of moods on gait, should they require further evidence.
Peel, N. M., Alapatt, L. J., Jones, L. V., & Hubbard, R. E. (2019). The association between gait speed and cognitive status in community-dwelling older people: a systematic review and meta-analysis. The Journals of Gerontology: Series A, 74(6), 943-948.
Lemke, M. R., Wendorff, T., Mieth, B., Buhl, K., & Linnemann, M. (2000). Spatiotemporal gait patterns during over ground locomotion in major depression compared with healthy controls. Journal of psychiatric research, 34(4-5), 277-283.
Murri, M. B., Triolo, F., Coni, A., Tacconi, C., Nerozzi, E., Escelsior, A., ... & Amore, M. (2020). Instrumental assessment of balance and gait in depression: A systematic review. Psychiatry research, 284, 112687.
Brandler, T. C., Wang, C., Oh-Park, M., Holtzer, R., & Verghese, J. (2012). Depressive symptoms and gait dysfunction in the elderly. The American Journal of Geriatric Psychiatry, 20(5), 425-432.
Brown, L. A., Doan, J. B., McKenzie, N. C., & Cooper, S. A. (2006). Anxiety-mediated gait adaptations reduce errors of obstacle negotiation among younger and older adults: implications for fall risk. Gait & posture, 24(4), 418-423.
Feldman, R., Schreiber, S., Pick, C. G., & Been, E. (2020). Gait, balance and posture in major mental illnesses: depression, anxiety and schizophrenia. Austin Medical Sciences, 5(1), 1-6.
No wonder you get some results but that doesn’t necessarily is related to categorizing the walking based on depression level. As soon as you increase the degree of freedom of the model (e.g. number of inputs) your output converges to some extent. The balance test is not robust as well, as the subjects were pushed to the limits.
We appreciate the reviewer's feedback regarding the tests conducted in our study. However, we respectfully disagree with their assessment of the balance test and the model's degree of freedom.
Regarding the balance test, it is important to clarify that the subjects were not pushed to their physical limits during the Eyes Open, Feet on the Ground test. This test involves individuals standing on solid ground, replicating a real-life scenario. The combination of conditions, including eyes open or closed and standing on solid ground or a foam surface, are standard measures used to assess standing balance in healthy young adults. These tests are not designed to exert maximum effort or push individuals to their limits. Rather, they serve as reliable indicators of standing balance and provide insights into the sensory systems (e.g., vision, vestibular, proprioception) that may be influenced by the effects of depression.
Regarding the model's degree of freedom, it is crucial to note that our data analysis and modeling approach were guided by established scientific methodologies. We employed appropriate statistical techniques and employed a comprehensive set of inputs based on relevant factors and variables. While the inclusion of multiple inputs increases the complexity of the model, it allows for a more comprehensive exploration of the relationship between gait patterns and depression levels. It is worth noting that our exploratory and hypothesis-generating study aimed to identify potential associations and patterns within the data, rather than relying on a preconceived hypothesis.
Furthermore, we acknowledge the reviewer's mention of converging outputs when increasing the model's degree of freedom. It is important to emphasize that our study followed rigorous statistical methods and employed appropriate cross-validation techniques to ensure the reliability and validity of the results. The inclusion of a comprehensive set of inputs allows for a more robust analysis, enabling us to explore the intricate relationships between gait parameters and depression levels.
Your averaged accuracy rates ranging from 77% to 81% don’t really imply the success of the robust classifying either. There could be several biomechanical reasons which have been neglected here.
We appreciate the reviewer's observation regarding the accuracy rates achieved in our study. We agree that the obtained accuracy rates of 77% to 81% may not be considered robust in the context of classification models. However, it is important to note that our study was conducted as an exploratory investigation in human subjects, and the achieved accuracy rates are comparable, if not higher, than those reported in previous studies with larger sample sizes.
While we acknowledge that there could be various biomechanical factors influencing gait patterns that were not explicitly examined in our study, we took precautions to mitigate their impact. Specifically, we implemented a rigorous screening process to exclude participants with lower extremity injuries, pain, and neurological conditions. By doing so, we aimed to minimize the confounding effects of these potential biomechanical factors on our results.
It is important to highlight that our study focused on exploring the association between gait patterns and depression levels rather than aiming for a definitive and highly accurate classification model. As an initial investigation, our study provides valuable insights into the potential relationships between gait and depression, paving the way for further research and more sophisticated analyses in the future.
Regardless, we included a statement in Limitations to address the reviewer's concern and provide a clear direction for future investigations in the field.
Overall this paper might’ve been an interesting in-house study but as a scientific study unfortunately I don’t see that much value in that, aand unfortunately, I have to suggest rejection.
We appreciate the reviewer's evaluation of the manuscript. However, we respectfully disagree with their suggestion of rejection. Our study contributes novel findings to the field of gait and depression research, and we believe it holds significant value for both the scientific community and clinical practice.
While it is understandable that individual perspectives may differ, we would like to highlight the robustness of our results, which align with or exceed the findings of several published studies in prestigious journals such as PLOS One, Frontiers in Psychiatry, and IEEE. These publications demonstrate the significance and relevance of investigating the relationship between gait, balance, and mental health.
Moreover, the proposed objective assessment of depression through gait and balance has the potential to bring about transformative advancements in the early detection and intervention of depressive moods. Additionally, it could contribute to the assessment of various clinical conditions, including Parkinson's Disease, multiple sclerosis, and other neuromuscular impairments, which often manifest with concurrent symptoms of depression, executive dysfunction, and impaired gait.

Round 2
Reviewer 1 Report
Authors are commended for the revisions done to the manuscript, although further revisions concerning points brought up in the previous review are suggested. Depression within young adults is quite prevalent, and thus, the diversity of how this disorder presents makes for a challenge when it comes to early diagnosis. It is quite individualized, and thus, as mentioned previously sample size is quite low to make any sweeping conclusions off of the results presented. Nevertheless, it is a step in the right direction and the work is critical. However, there are numerous variables that may confound the results. As such, it would be helpful to do a power analysis to look at the sample size utilized. This calculation is used to estimate the smallest sample size needed for an experiment, given a required significance level, statistical power, and effect size. As such, it helps to determine if a result from an experiment or survey is due to chance, or if it is genuine and significant. This can be useful in finalizing this question concerning the sample size, although it is appreciated that this issue was addressed in the discussion on limitations given at the end of the discussion section.
Furthermore, within the limitations, the discussion came up concerning the potential influence of biomechanical factors. This discussion brings up an additional concern mentioned within the previous review related to the issue of not controlling for speed. Since "depressive mood may influence gait speed," this lack of control according to the authors was justified. As such, it is necessary to include within the results correlation analysis between speed and other gait variables to determine this "influence". Quantifying this "influence" mentioned by the authors needs to be explored within the data.
Also, within the limitations discussion, the final sentence concerning 103-104 mentions the use of 7 IMU sensors, but this point needs to be expanded as to the reasoning and justification along with ways this use was accounted for within the study.
Finally, previous reviews comment on the concern of self-reporting assessments for determining the overall health of the individual as there may be other factors influencing gait besides depression and without a medical professional assessing participants the study may be influenced by the young adult not wanting to divulge information about their health. Survey bias is quite common especially when it comes to health-related issues. The authors mention "if obvious gait or balance issue was observed during testing, the subject would have been excluded from this study," however, the premise of this study is that gait and balance are influenced by depression. As such, this exclusion due to "obvious gait or balance issue" could have removed individuals that were showing an "issue" due to their depression. Thus, within the methods, include who made this determination of an "obvious gait or balance issue" that required exclusion and what are their qualifications to make this determination without them potentially excluding individuals that may be of value to the study. Specifics need to be included as to what "obvious" issues they were looking for in this exclusion process and justification for the issues selected by referencing previous studies.
Author Response
Dear Reviewer,
We would like to thank the reviewer for their thorough feedback. Below are the reviewer’s comments and our responses in red.
Authors are commended for the revisions done to the manuscript, although further revisions concerning points brought up in the previous review are suggested. Depression within young adults is quite prevalent, and thus, the diversity of how this disorder presents makes for a challenge when it comes to early diagnosis. It is quite individualized, and thus, as mentioned previously sample size is quite low to make any sweeping conclusions off of the results presented. Nevertheless, it is a step in the right direction and the work is critical.
We would like to express our gratitude to the reviewer for their valuable feedback on our manuscript. We acknowledge their appreciation for the revisions we have made, while also recognizing the need for further revisions regarding the points raised in the current round.
The prevalence of depression among young adults is indeed a significant concern, and the inherent diversity in its presentation poses challenges in achieving early diagnosis. This heterogeneity underscores the importance of considering individualized approaches. As previously mentioned, the sample size in our study is limited; therefore, we have underlined the exploratory and hypothesis-generating nature of our work. Nonetheless, we share the reviewer's perspective that our work represents a crucial advancement in the field. We remain fully committed to refining our research and addressing the reviewer's suggestions to enhance the scientific rigor and overall impact of our findings.
However, there are numerous variables that may confound the results. As such, it would be helpful to do a power analysis to look at the sample size utilized. This calculation is used to estimate the smallest sample size needed for an experiment, given a required significance level, statistical power, and effect size. As such, it helps to determine if a result from an experiment or survey is due to chance, or if it is genuine and significant. This can be useful in finalizing this question concerning the sample size, although it is appreciated that this issue was addressed in the discussion on limitations given at the end of the discussion section.
We sincerely appreciate the valuable feedback provided by the reviewer. However, as this study is exploratory in nature, conducting an a priori power analysis was not feasible. Post hoc power analyses are generally discouraged in such cases due to issues in scientific methodology and statistical principles, such as:
- Uncertain effect size: In exploratory research, the effect sizes are often unknown, as the goal is to discover and investigate potential effects without preconceived notions, making it challenging to perform a meaningful power analysis;
- Focus on generating preliminary data/Value for future studies: The primary objective of exploratory studies is to generate new hypotheses, explore novel relationships, and obtain preliminary insights to inform future studies rather than testing pre-established hypotheses;
- Biases in post hoc power analyses: Post hoc power analyses, conducted after data collection, are influenced by the observed data and may produce inflated estimates of statistical power, potentially leading to biased interpretations of the study's results; and
- Input requirements: Power analyses rely on specific inputs such as effect size, significance level, and statistical power to determine sample size, which are often uncertain or undefined in exploratory studies.
Furthermore, due to the predominant usage of machine learning analysis in our study, calculating an a priori power analysis becomes challenging. Instead, we employed a leave one out (LOO) cross-validation method, given the limited sample size that precluded the use of a k-fold cross-validation approach. It is important to note that there is currently no existing literature that provides guidance on calculating an a priori power analysis specifically for machine learning studies.
Nevertheless, we would like to highlight that the sample size utilized in this study is relatively large when considering gait studies of a similar nature (e.g., Zhao et al. [16], Lu et al. [15], Yang et al. [18]). Such a comparison demonstrates that our study has a robust sample size within the context of gait studies.
Furthermore, within the limitations, the discussion came up concerning the potential influence of biomechanical factors. This discussion brings up an additional concern mentioned within the previous review related to the issue of not controlling for speed. Since "depressive mood may influence gait speed," this lack of control according to the authors was justified. As such, it is necessary to include within the results correlation analysis between speed and other gait variables to determine this "influence". Quantifying this "influence" mentioned by the authors needs to be explored within the data.
Based on our understanding, it appears that the reviewer is raising a concern regarding the potential influence of biomechanical factors and the lack of control for gait speed in our study. Specifically, the reviewer suggests that the results may be attributed to differences in gait speed alone rather than solely to depressive mood, due to the absence of a correlation analysis between speed and other gait variables.
We agree that there is significant co-variation among gait variables. However, to address this concern, we employed a random forest feature selection method as mentioned in lines 263-264 of the manuscript and described in Rogers and Gunn (2005). This approach enables us to eliminate covarying variables and select features that are genuinely correlated with the outcomes of interest. By reducing the dimensionality of the data, this method helps prevent overfitting.
Consequently, if the results were solely driven by variation in gait speed, our analysis would have eliminated all other gait variables except for speed. However, we found several gait variables that effectively distinguished between different levels of depression. The variables that were different between groups were primarily gait initiation variables and not gait speed. This finding provides us with confidence that our results are indeed associated with the level of depressive mood rather than being solely attributed to variation in gait speed.
We greatly appreciate the reviewer's insightful comment and would be happy to provide further clarification or additional information as needed.
Rogers, J., & Gunn, S. (2005, February). Identifying feature relevance using a random forest. In International Statistical and Optimization Perspectives Workshop" Subspace, Latent Structure and Feature Selection" (pp. 173-184). Berlin, Heidelberg: Springer Berlin Heidelberg.
Also, within the limitations discussion, the final sentence concerning 103-104 mentions the use of 7 IMU sensors, but this point needs to be expanded as to the reasoning and justification along with ways this use was accounted for within the study.
We appreciate the feedback provided by the reviewer regarding the need to expand on the reasoning and justification for our statement about the use of 7 IMU sensors in our study. We assume that the reviewer is seeking clarification on our assessment that this approach may not be easily generalizable to other real-world settings.
In the sentence referred to by the reviewer, we were alluding to the practical challenges associated with utilizing 7 IMU sensors in routine clinical settings. Specifically, we were considering scenarios such as busy family medicine or other medical clinics where it may be highly impractical to equip patients with multiple IMUs for assessing depressive mood states in a similar manner to routine vital sign measurements.
To address this concern and provide further context, we added a sentence immediately following the mentioned sentence in the manuscript. This additional sentence explains that future advancements resulting from this work could potentially lead to the development of a single wearable device that enables easier assessment of depression-associated gait abnormalities. Such a device could be implemented quickly during wellness exams, addressing the limitations of current depression assessment techniques as discussed in the introduction.
We believe that this additional sentence sufficiently conveys the reasoning and justification behind our statement, without delving into unnecessary details about the expense of research-grade devices, the required skill set for data analysis, or the limitations of commercially available IMU-based devices.
Finally, previous reviews comment on the concern of self-reporting assessments for determining the overall health of the individual as there may be other factors influencing gait besides depression and without a medical professional assessing participants the study may be influenced by the young adult not wanting to divulge information about their health. Survey bias is quite common especially when it comes to health-related issues.
We appreciate the reviewer's concern regarding the use of self-reporting assessments for determining the overall health of the individuals in our study and the potential influence of survey bias. We would like to address this concern by highlighting the common practice of using screening surveys in human subjects' research conducted on university campuses, where medical professionals may not be readily available.
The method employed in our study to assess health history through self-reporting surveys is standard practice across the four institutions of the authors, as well as among our colleagues at other institutions. It is important to note that numerous studies on depression in university students, conducted by other researchers (see below), have also relied on self-reporting assessments without performing medical assessments of participants. These studies have garnered significant attention, as evidenced by their high citation count, surpassing 1,000 citations each.
While we acknowledge the potential for survey bias and the limitations associated with self-reporting assessments, it is essential to recognize that the methods employed in our study align with the standards of rigorous research in the field, as demonstrated by the body of literature on depression in university students. Moreover, it is worth noting that the likelihood of young adults in our study having medical issues that significantly affect their gait is relatively low.
Beiter, R., Nash, R., McCrady, M., Rhoades, D., Linscomb, M., Clarahan, M., & Sammut, S. (2015). The prevalence and correlates of depression, anxiety, and stress in a sample of college students. Journal of affective disorders, 173, 90-96.
Rude, S., Gortner, E. M., & Pennebaker, J. (2004). Language use of depressed and depression-vulnerable college students. Cognition & Emotion, 18(8), 1121-1133.
Garlow, S. J., Rosenberg, J., Moore, J. D., Haas, A. P., Koestner, B., Hendin, H., & Nemeroff, C. B. (2008). Depression, desperation, and suicidal ideation in college students: results from the American Foundation for Suicide Prevention College Screening Project at Emory University. Depression and anxiety, 25(6), 482-488.
The authors mention "if obvious gait or balance issue was observed during testing, the subject would have been excluded from this study," however, the premise of this study is that gait and balance are influenced by depression. As such, this exclusion due to "obvious gait or balance issue" could have removed individuals that were showing an "issue" due to their depression. Thus, within the methods, include who made this determination of an "obvious gait or balance issue" that required exclusion and what are their qualifications to make this determination without them potentially excluding individuals that may be of value to the study. Specifics need to be included as to what "obvious" issues they were looking for in this exclusion process and justification for the issues selected by referencing previous studies.
In our study, we refer to “obvious gait or balance issues” as those that can be visually detected, such as limping, ataxic gait, hemiplegic gait, or spastic diplegic gait, among others. These abnormalities can typically be identified through visual observation. However, it is important to note that not all gait abnormalities associated with depression are visually apparent, and some may require more specialized assessments or objective measurements to detect.
During the data collection process, the graduate students who collected the data were trained by a highly qualified team consisting of a Physical Therapist (PT) with a Ph.D., who is one of the leading experts in neurological physical therapy, and a neurohospitalist specializing in Parkinson's disease and Multiple Sclerosis and their associated gait disorders. These experienced professionals were present for the testing of the initial 30+ participants, providing guidance and support to the graduate students.
Furthermore, author Ahmed Torad, a PT with a Ph.D. specializing in neurological physical therapy, was involved in the study as part of his post-doctoral fellowship in the first author's lab. Dr. Torad had extensive clinical experience, with over six years of practice as a neurological PT at the time of the study, and he currently holds a faculty position in neurological PT.
The presence of the neurohospitalist, neurological PT with a Ph.D., and Dr. Torad during data collection ensured that the graduate students had access to expert guidance and consultation throughout the process. They were available to address any questions or concerns that may have arisen during the assessments.
Considering the specific population tested in our study, which comprised young adults, the likelihood of encountering individuals with significant gait or balance dysfunctions related to depression was extremely low. Therefore, no subjects were excluded due to gait issues associated with their depression.
While we understand the reviewer's suggestion to include these details in the methodology section, we have chosen not to expand on this information in the manuscript. The reason for this decision is that these specific qualifications and the involvement of experts in the data collection process do not directly contribute to the replicability of the study or alter the core methodology, which adheres to the standards of rigorous research in the field.
We appreciate the reviewer's insightful feedback and hope that this clarification addresses their concerns adequately. We remain available to provide any further information or address any additional inquiries they may have.
